# Characterisation of a Novel Insect-Specific Virus Discovered in Rice Thrips, *Haplothrips aculeatus*

**DOI:** 10.3390/insects15050303

**Published:** 2024-04-24

**Authors:** Hao Hong, Zhuangxin Ye, Gang Lu, Kehui Feng, Mei Zhang, Xiaohui Sun, Zhilei Han, Shanshan Jiang, Bin Wu, Xiao Yin, Shuai Xu, Junmin Li, Xiangqi Xin

**Affiliations:** 1Shandong Key Laboratory of Plant Virology, Institute of Plant Protection, Shandong Academy of Agricultural Sciences, Jinan 250100, China; honghao_ahjx@163.com (H.H.); xsh1122a@163.com (S.X.); 2State Key Laboratory for Managing Biotic and Chemical Threats to the Quality and Safety of Agro-Products, Key Laboratory of Biotechnology in Plant Protection of Ministry of Agriculture and Zhejiang Province, Institute of Plant Virology, Ningbo University, Ningbo 315211, China

**Keywords:** *Haplothrips aculeatus*, thrips, metatranscriptomic sequencing, *Ollusvirus*, negative-strand RNA virus, RdRp

## Abstract

**Simple Summary:**

Insect-specific viruses (ISVs) are increasingly recognised for their role in causing severe illnesses and mortality in both humans and animals. In this study, the full genome of ISV (Rice thrips ollusvirus 1, RTOV1) was revealed in rice thrips using metatranscriptome sequencing, RT-PCR, and RACE, respectively. RTOV1 has a typical linear G-N-L genome structure of the order *Jingchuvirales*. Phylogenetic analysis categorised this virus as an *Ollusvirus*. The infection of host insects with RTOV1 triggered antiviral RNA interference (RNAi), resulting in a significant accumulation of 22-nt virus-derived small interfering RNAs (vsiRNAs), with a notable bias towards the A/U content. Our study provides valuable information on ISVs in thrips that may be useful for pest management.

**Abstract:**

Insects constitute the largest proportion of animals on Earth and act as significant reservoirs and vectors in disease transmission. Rice thrips (*Haplothrips aculeatus,* family Phlaeothripidae) are one of the most common pests in agriculture. In this study, the full genome sequence of a novel *Ollusvirus*, provisionally named “Rice thrips ollusvirus 1” (RTOV1), was elucidated using transcriptome sequencing and the rapid amplification of cDNA ends (RACE). A homology search and phylogenetic tree analysis revealed that the newly identified virus is a member of the family *Aliusviridae* (order *Jingchuvirales*). The genome of RTOV1 contains four predicted open reading frames (ORFs), including a polymerase protein (L, 7590 nt), a glycoprotein (G, 4206 nt), a nucleocapsid protein (N, 2415 nt) and a small protein of unknown function (291 nt). All of the ORFs are encoded by the complementary genome, suggesting that the virus is a negative-stranded RNA virus. Phylogenetic analysis using polymerase sequences suggested that RTOV1 was closely related to ollusvirus 1. Deep small RNA sequencing analysis reveals a significant accumulation of small RNAs derived from RTOV1, indicating that the virus replicated in the insect. According to our understanding, this is the first report of an *Ollusvirus* identified in a member of the insect family Phlaeothripidae. The characterisation and discovery of RTOV1 is a significant contribution to the understanding of *Ollusvirus* diversity in insects.

## 1. Introduction

An increasing number of novel ISVs are being discovered using next-generation sequencing (NGS) technology and transcriptome analysis [1,2,3]. The identification of a large number of viral sequences has led to the realisation that the presence of viruses in ecosystems is ecologically and evolutionarily important [4,5]. Some insects have been shown to serve as viral vectors for animals or plants, and the viruses in insects mainly include insect-specific viruses (ISVs), vector-borne pathogenic viruses, and viruses from insect endosymbiotic microorganisms or digestive matter [6,7].

Negative-strand RNA viruses (NSVs) cause various diseases in plants, animals and humans and are broadly classified into segmented and non-segmented viruses [8]. An increasing number of plant- and insect-infecting NSVs are being discovered, and they are considered emerging pathogens. The new NSVs in the family of *Aliusviridae* belong to the order *Jingchuvirale*, which was created by the International Committee on Classification of Viruses (ICTV) in 2018 [9,10]. The order *Jingchuvirale* presents a diverse array of genome organisations, encompassing unsegmented, bi-segmented and circular configurations [11]. The viral genomes encode a glycoprotein (G), a nucleoprotein (N) and a polymerase (L), though some viral genomes may lack the glycoprotein. It has been reported that the loss of the G gene may have occurred during the virus’s long-term evolution [1]. *Ollusvirus* belongs to the family *Aliusviridae*, with a genome structure of G-N-L, and it is a genus with relatively few viruses [10].

A growing body of research suggests that arthropods may be important reservoirs for a wide range of viruses and may play an important role in virus evolution [1,12]. *Haplothrips aculeatus*, belonging to the family Phlaeothripidae and order Thysanoptera, is a rice pest that directly feeds on rice and can transmit plant viruses, causing significant damage to agricultural production [13,14]. *H. aculeatus* is distributed in all the rice-growing areas of Asia and in most of China. It is widely parasitic on cereal crops and a variety of grass weeds. To further evaluate the potential causes of rice losses caused by *H. aculeatus*, we analysed the possible insect-transmitted pathogens using RNA-seq. In this paper, we have successfully identified a new ISV from *H. aculeatus* and classified it as the genus *Ollusvirus* in the family *Aliusviridae*. This is of significant importance for expanding our understanding of viruses and provides a theoretical basis for future research on whether the disease caused by this virus will become prevalent.

## 2. Materials and Methods

### 2.1. Sample Preparation and RNA Extraction

Insect samples of *H. aculeatus* were collected from a rice field in Shandong, China, in July 2021. The extraction of total RNA proceeded as previously described, with some modifications for optimisation [15]. Specifically, each sample consisted of 50 adult thrips, and RNA extraction was performed using AG RNAex Pro Reagent (agbio, Changsha, Hunan, China) instead of Trizol reagent (Invitrogen, Carlsbad, CA, USA). The methods used were exactly the same as previously described. The quality of the RNA was confirmed using a NanoDrop spectrophotometer (Thermo Scientific, Waltham, MA, USA). The cDNA synthesis for whole-genome sequence determination, transcriptomic sequencing and small RNA (sRNA) sequencing, respectively, used 2 μg of RNA each, while 500 ng of RNA was used for first-strand cDNA synthesis in RACE.

### 2.2. Host Insect Identification

The assembled contigs were first compared with the Barcode of Life Data Systems (accessed on 14 December 2019, https://www.boldsystems.org/) to determine the mitochondrial cytochrome coxidase subunit I (COI) sequence of the rice thrips species. The COI sequence was then confirmed via a BLASTn search against the nucleotide database of the National Center for Biotechnology Information (NCBI).

### 2.3. Transcriptomic and Small RNA (sRNA) Sequencing

Novogene (Tianjin, China) conducted transcriptomic and small RNA (sRNA) sequencing on the aforementioned insect RNA samples. The transcriptome and sRNA libraries were prepared as described previously [16]. In brief, sequencing was performed on the constructed paired-end (150 bp) libraries using the Illumina HiSeq 4000 platform. The raw reads were quality trimmed using Trimmomatic software (version 0.39), and subsequently de novo assembled [17]. The Illumina TruSeq Small RNA Sample Preparation Kit (Illumina, USA) was used to prepare the sRNA libraries for sequencing. Subsequently, sRNA sequencing was carried out on the Illumina HiSeq 2500 platform. The Cutadapt tool was then used to remove the adapters and low-quality sequences from the raw output data [18].

### 2.4. Virus Discovery and Confirmation by Reverse Transcription-PCR (RT-PCR)

The identification of the viral contig involved the following steps: Initially, the assembled contigs were aligned against a locally downloaded virus database sourced from the NCBI viral reference database (accessed on 9 July 2023, https://www.ncbi.nlm.nih.gov/genome/viruses). Afterward, in order to prevent false positives, the putative viral sequence was subjected to comparison against both the NCBI nucleotide (NT) and non-redundant (NR) protein databases. Finally, the identified viral contigs were validated using RT-PCR, followed by Sanger sequencing using specific primers (Appendix A).

### 2.5. Determination of Viral Genome Termini and Transcript Abundance

The full genome sequence of the virus was obtained by rapid amplification using cDNA ends (RACE) technology. Briefly, the 5′-RACE and 3′-RACE cDNA were synthesised using the 5′/3′ RACE kit (agbio, Changsha, China) according to the manufacturer’s instructions. Subsequently, the PCR products were cloned into the pTO vector (Generalbiol, Chuzhou, Anhui, China), and Sanger sequencing was performed on the positive recombinant plasmids. The primers utilised for RACE are detailed in Appendix A.

To assess the transcript coverage and abundance of the viruses, we utilised both Bowtie2 and Samtools to align the adaptor-trimmed and quality-trimmed reads of the transcriptome back to the complete viral genomes [19,20]. Subsequently, the coverage of the aligned reads on the viral genomes was visualised using the Integrated Genomics Viewer (IGV) [21].

### 2.6. Small RNA Analysis

Small RNA analysis was carried out as described previously [22]. First, an sRNA library was prepared using the Illumina TruSeq sRNA Sample Preparation Kit (Illumina, San Diego, USA) and sequenced on the Illumina HiSeq 2500 platform. Next, the raw data were processed to obtain clean reads, from which sRNAs with lengths ranging from 18 to 30 nucleotides were extracted for further analysis. Finally, the sRNA reads from the previous step were mapped back to the entire viral genome sequence using Bowtie (allowing for zero mismatches) to identify vsiRNAs. The results were then further analysed using custom Perl scripts and Linux Bash scripts to analyse the obtained vsiRNAs [23].

### 2.7. Genome Annotation and Phylogenetic Analysis

The ORF Finder program at the NCBI (accessed on 20 January 2024, https://www.ncbi.nlm.nih.gov/orffinder/) was utilised to predict potential open reading frames (ORFs) within the virus genomes. The conserved protein structural domains were predicted using InterProScan (version InterPro 98.0, https://www.ebi.ac.uk/interpro; accessed on 20 January 2024). The amino acid sequence of the predicted *Ollusvirus* RdRp was obtained from the NCBI. Sequences were aligned using MAFFT (version 7.450) with Gblock [24]. The alternative model was evaluated using ModelTest-NG according to the default parameters [25]. Subsequently, maximum likelihood (ML) trees were constructed using RAxMLNG (version 0.9.0) with 1000 bootstrap replications [26].

## 3. Results

### 3.1. Discovery of RNA Virus-Related Sequences in H. aculeatus

To identify the RNA virus-related sequences in *H. aculeatus*, NGS was used to analyse the total RNA samples by RNA sequencing. The results show that 13.08 Gbp of the raw data from the cDNA library of insect samples were obtained by deep sequencing using the Illumina HiSeq 4000 platform (150 bp paired-end reads) of Novogene. A total of 23,001,625 clean reads were obtained. The assembled contigs were first compared with the Barcode of Life Data Systems to determine the cytochrome coxidase subunit I (COI) sequence of the rice thrips species. Then, the COI sequence was confirmed by a BLASTn search against the nucleotide database of the National Center for Biotechnology Information (NCBI). The result showed that the COI sequence was 99.66% identical to that of *H. aculeatus* (GenBank: MF716896.1) (Appendix A). A total of 2054 virus-like reads were found in this thrips RNA library. To identify the viral contig, the assembled contigs were searched against the NCBI viral reference database. The result revealed the presence of a new RNA virus belonging to the genus *Ollusvirus*. The viral sequence was validated using RT-PCR, and the complete genome sequence of *Ollusvirus* was obtained by RACE.

### 3.2. Characterisation of Rice Thrips Ollusvirus 1

The full genome sequence of RTOV1 is 16,282 nt long, including a 223 nt 5′ untranslated region (UTR), with four non-overlapping predicted ORFs (ORF1, 224–7813 nt; ORF2, 8148–10,562 nt; ORF3, 12,018–16,223 nt; ORF4, 12,018–16,223 nt), and a 59 nt 3′ UTR (Figure 1A). Based on the InterProScan prediction of conserved domains for RTOV1, ORF4 contains a Mononegavirales RNA-directed RNA polymerase catalytic domain (Mo-noneg_RNA_pol_cat) and a Mononegavirales mRNA-capping domain V (Mo-noneg_mRNAcap) (Figure 1A). To characterise the proteins encoded by these ORFs, we used the NCBI Open Reading Frame Finder. Three of these ORFs encode major viral proteins. ORF1 was predicted to encode a 155.32 kDa glycoprotein (Gp), ORF3 was predicted to encode an 88.33 kDa nucleoprotein (N), and ORF4 was predicted to encode a 289.1 kDa polymerase protein (RdRp) (Appendix A). However, no biological function could be attributed to ORF2 on the basis of protein homology (Figure 1A). The virus was tentatively named “Rice thrips ollusvirus 1” (RTOV1), and the full genome sequence was submitted to the GenBank database (GenBank: OR886228) (Appendix A). The genome structure of RTOV1 was highly similar to that of Atractomorpha sinensis ollusvirus 1 (ASOV1, genus: *Ollusvirus*), which was previously detected by in Pink-Winged Grasshopper, *Atractomorpha sinensis* (genus: Atractomorpha; family: Pyrgomorphidae) (Figure 1B) [27].

RTOV1 shared 21.61% identity in the capsid protein (CP, also named nucleoprotein in NSVs) region with its closest homologue, ollusvirus 1 (ASOV1, GenBank: WAB51682.1), meeting the delimitation criterion for establishing a new species (less than 90% CP identity). The abundance and coverage of RTOV1 were assessed by realigning RNA-seq reads with the full genome sequence of RTOV1 obtained. A total of 22,701,157 reads were perfectly aligned to the RTOV1 genome, representing 0.0048% of the total RNA-seq reads (Figure 1A). As shown in Figure 1A, the transcripts were distributed across the viral genome, with a particularly high level of abundance at the 3′ terminus, indicating that RTOV1 efficiently replicated in the insects.

### 3.3. Phylogenetic Analysis of RTOV1 and Related Jingchuvirals

For a full insight into the evolution of RTOV1, phylogenetic trees of the viral Gp, N and RdRp proteins were generated, respectively. As illustrated in Figure 2A, RTOV1 RdRp is on a branch with ollusvirus 1 of the genus *Ollusvirus* in a phylogenetic tree based on the jingchuviral RdRp protein sequences. The N protein tree indicated that RTOV1 was related to ollusvirus 1 (Figure 2B). Similarly, the RTOV1 GP was found to be closely related to ollusvirus 1 (Figure 2C).

In addition, a BLASTp search of the NCBI reference viral sequence database was performed to identify the viruses related to RTOV1. The results showed that the ORF1 (Gp) sequence shared the highest sequence similarity with Beetle aliusvirus (GenBank: WPR17560.1) of the family *Aliusviridae* with 21.74% amino acid sequence identity. With 21.61% amino acid sequence identity, the N protein sequence showed the highest sequence similarity to ollusvirus 1. Similarly, the RTOV1 RdRp protein sequences exhibited the highest sequence similarity to Osmia-associated bee chuvirus (OABV-49, also named *Chuviridae* sp. GenBank: BDG58444.1), a novel *Ollusvirus* identified in wild bees, *O. taurus* (32.09% amino acid sequence identity) (Appendix A) [28].

To investigate the conserved motifs of the viral RdRp, we identified five conserved motifs in the RdRp sequences of RTOV1 and the four homologous Ollusviruses using the default parameters of MEME (Version 5.5.5, https://meme-suite.org/meme/tools/meme; accessed on 20 January 2024) (Appendix A). In conclusion, the aforementioned results suggest that the newly identified RTOV1 could be added to the family *Aliusviridae*.

### 3.4. Activation of Antiviral RNA Interference Pathway in H. aculeatus Responsive to ISVs

RNA silencing serves as a crucial antiviral immune response in insects, playing a significant role in the elimination of viruses [29,30]. Viral dsRNA is cleaved by the insect RNase III enzyme Dicer-2 (Dcr2), and subsequently generates large amounts of virus-derived siRNAs (vsiRNAs) in the host [31]. To investigate siRNA-based antiviral immunity in response to the RTOV1 infection of *H. aculeatus*, small RNAs (sRNAs) of the host insect were sequenced, and the vsiRNAs were comprehensively analysed. A total of 2,699,871 vsiRNA reads, including 1350 unique reads, were perfectly mapped to the assembled RTOV1 genome sequence (Figure 3A). Of these vsiRNAs, more were derived from the positive-sense strand than from the negative-sense strand of the viral genome (Figure 3B). To provide further analyses of the types of these vsiRNAs, it was shown that vsiRNAs from the viral positive-sense strand of the viral genome were predominantly 22 nt in length (Figure 3B).

The vsiRNAs displayed a clear preference for A/U at their 5′-terminal nucleotides and were evenly spread throughout the viral genome (Figure 3A,C). Notably, although the RTOV1 vsiRNAs were distributed throughout the viral genome, there was an extremely high abundance at the 5′-terminal nucleotides (Figure 3A). These results suggest that RTOV1 replicated in *H. aculeatus*, and viral RNA was cleaved by the host antiviral RNA interference pathway into predominantly 22 nt-sized small RNAs.

## 4. Discussion

Many RNA viruses have been discovered through the application of HTS and sophisticated bioinformatics techniques, leading to an enhanced comprehension of insect viromes and the evolutionary dynamics of viruses. As an important agricultural pest, thrips can transmit multiple viruses, leading to the occurrence of viral diseases. Western flower thrips, *Frankliniella occidentalis* Pergande, primarily cause the spread of viruses belonging to the genus *Orthotospovirus* (e.g., Tomato spotted wilt virus and Tomato chlorotic spot virus) and Tobacco streak virus (TSV; genus *Ilarvirus*) [32,33]. *Thrips palm* Karny can cause the harm of viruses like Watermelon silver mottle virus (WSMoV; genus *Orthotospovirus*), Peanut bud necrosis virus (PBNV; genus *Orthotospovirus*) and Muskmelon yellow spot virus (MYSV; genus *Orthotospovirus*) [34,35]. *Thrips tabaci* Lindeman play a crucial role in the prevalence of diseases, such as Tobacco ringspot virus (TRSV; genus *Nepovirus*, family *Secoviridae*), Sowbane mosaic virus (SoMV; genus *Sobemovirus*) and Iris yellow spot virus (IYSV; genus *Orthotospovirus*) [36,37,38]. All the thrips species that transmit plant viruses belong to the family Thripidae and the subfamily Thripinae. In this paper, we identified a new virus belonging to the genus *Ollusvirus* in family Phlaeothripidae and named it RTOV1. At present, 15 species of thrips have been confirmed to transmit viruses, accounting for just 0.2% of the total number of recorded thrips species [39]. In the future, investigating its virulence in the host, host range, and potential transmission to rice would not only deepen our understanding of the diversity and evolution of ISVs in insects, but also have significant implications for pest management strategies. Additionally, this would help uncover the potential reasons behind the damage caused to rice by thrip species.

The N, Gp and RdRp proteins are conserved throughout negative-sense RNA viruses, except for a few viruses where the Gp protein is lost during long-term evolution [1]. From three phylogenetic tree of these proteins, both of them show close phylogenetic relationships with ollusvirus 1. However, according to the results obtained from BLASTp analysis, only the N protein exhibited more homology with ollusvirus 1. The Gp and RdRp proteins showed strong homology with Beetle aliusvirus and Chuviridae sp, respectively. Due to a recently discovered ISV, there are a few research reports on Beetle aliusvirus [40]. According to the information provided in the GenBank database, this virus belongs to the order *Jingchuvirales* and the family *Aliusviridae*. Chuviridae sp, also known as Osmia-associated bee chuvirus (OABV-49), is an ISV belonging to the family *Aliusviridae* and the genus *Ollusvirus* [27]. According to the BLASTp data, we added this novel virus into the family *Aliusviridae*. Considering the evolutionary tree relationships, we further categorised this new virus into the genus *Ollusvirus*. Notably, the RdRp protein demonstrated a strong phylogenetic affinity and homology with Ollusviruses.

The abundance of virus-induced small RNAs intuitively reflects the strength of the host’s antiviral RNA interference response and, to some extent, indicates the level of viral proliferation in the host. The typical characteristics of vsiRNAs strongly suggest that the host antiviral RNAi plays an active role in responding to RTOV1 infection. More vsiRNAs originate from the positive strand rather than the negative strand (Figure 3B). Such results are likely due to the negative-sense RNA virus genome (minus strand) being protected by the N protein, thereby reducing cleavage by the RNAi pathway. The complementary strand (plus strand), however, may be extensively cleaved due to the lack of protection [41]. Interestingly, the vsiRNAs of TSWV generated from *Frankliniella fusca* (family Thripidae) show a higher abundance of 22 nt sRNAs compared to 21 nt [42]. As shown in Figure 3B, we also observed a higher quantity of 22 nt vsiRNAs from RTOV1 compared to 21 nt vsiRNAs. Although *H. aculeatus* and *F. fusca* do not belong to the same family, they both seem to mediate the abundant production of 22 nt vsiRNAs. We speculate that this may be due to a specific RNA cleavage mechanism in thrips.

## 5. Conclusions

In conclusion, a novel *Ollusvirus* was identified in *H. aculeatu*. The full-length genome of RTOV1 was revealed using metatranscriptome sequencing and RACE technology. To the best of our knowledge and the consulted literature, this is the first Ollusvirus identified in family Phlaeothripidae [3]. RTOV1 is 16,282 nt long and encodes three major viral proteins (RdRp, N and Gp). The host’s antiviral RNAi mechanism can target this virus, leading to the accumulation of abundant 21–24 nt vsiRNAs. The discovery of this new *Ollusvirus* not only enriches our understanding of insect microbial composition, but also helps reveal the diversity, evolution and ecological significance of insect viruses, which are crucial for agricultural production and ecosystem health.

## Figures and Tables

**Figure 1 insects-15-00303-f001:**
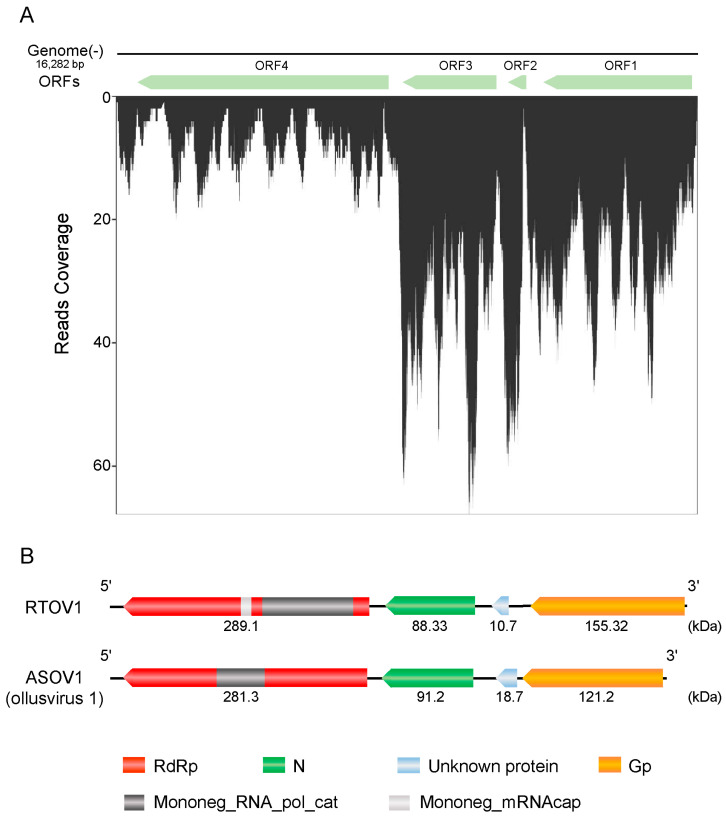
Characterisation of Rice thrips ollusvirus 1 (RTOV1). (**A**) High-quality reads were mapped against the validated, near full-length genome sequence of RTOV1. Predicted ORFs were manually annotated (green). The readmap depicts the nucleotide coverage by viral genomes RNA reads. Note the relatively higher RNA fraction towards the 3′ end of the genome. (**B**) Comparison of the genome organisation (not to scale) of RTOV1 and its closest relative ollusvirus 1 (ASOV1). The genomes of RTOV1 and ASOV1 are unsegmented negative-sense RNA, and RdRp (L), nucleoprotein (N), ORF2 (unknown biological function) and glycoprotein (Gp) have similar positions and mass ranges.

**Figure 2 insects-15-00303-f002:**
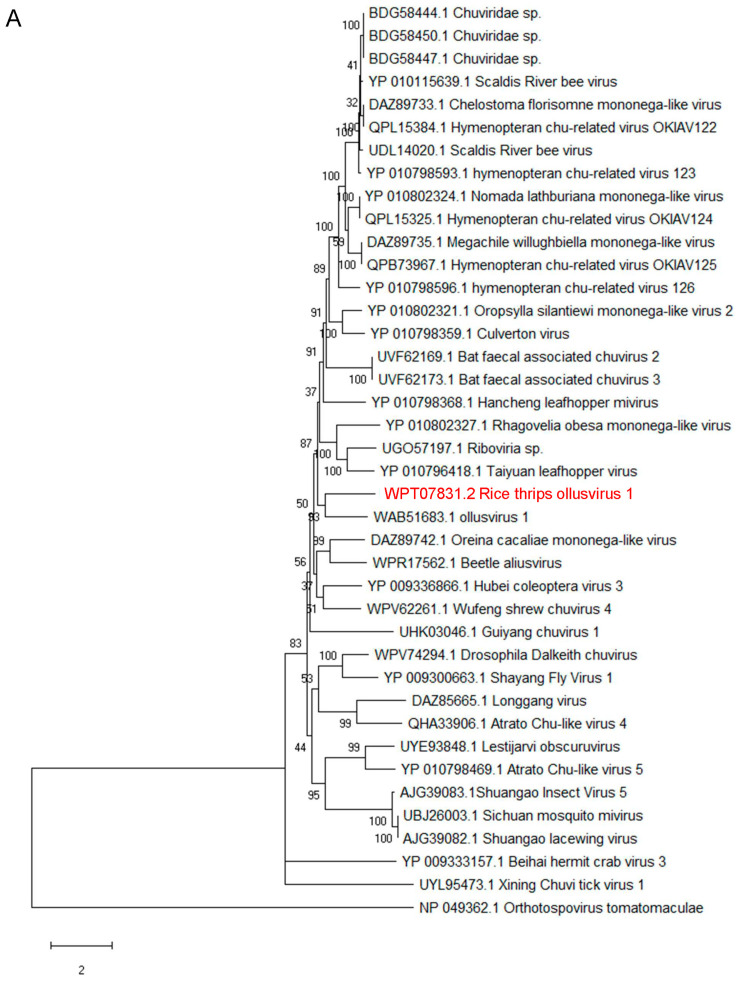
Phylogenetic analysis of RTOV1 and related *jingchuvirals* based on the putative amino sequences for RdRp (**A**), nucleoproteins (**B**) and glycoproteins (**C**). Phylogenetic trees were constructed using the maximum likelihood method, with bootstrap values calculated for 1000 replicates. The best-fit models were the LG (Le and Gascual) + G (gamma distributed) + F (Fregs.) model for the RdRp and nucleoprotein tree and the WAG (Whelan and Goldman) + G + F model for the glycoprotein tree. The RTOV1 isolate identified in this study is indicated by red text. RdRp, nucleoproteins and glycoproteins of a typical negative-sense RNA virus, Tomato spotted wilt virus (TSWV), were used as the outgroups. The scale bars represent percentage divergence.

**Figure 3 insects-15-00303-f003:**
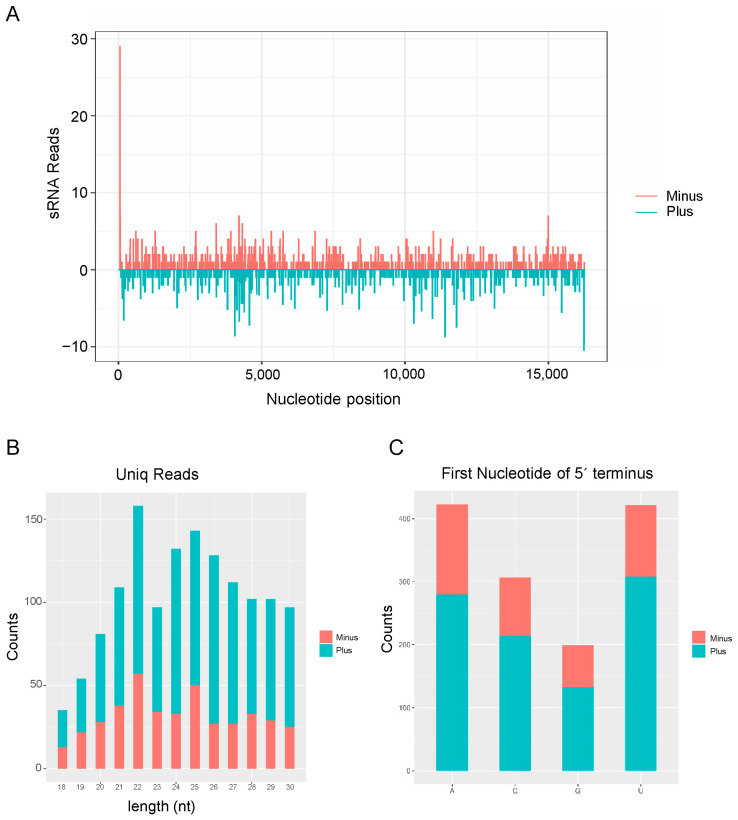
Virus-derived small interfering RNAs (vsiRNAs) of RTOV1. (**A**) Distribution of RTOV1 siRNAs alongside the viral genome. Cyan and pink represent siRNAs derived from the sense (plus strand) and antisense (minus strand) genomic strands of RTOV1, respectively. (**B**) Size distribution of RTOV1 siRNAs. (**C**) Five’-terminal nucleotide percentage of RTOV1 siRNAs.

## Data Availability

The data presented in this study are available within the article and Appendix A.

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
