# Peer review of "Characterisation of a Novel Insect-Specific Virus Discovered in Rice Thrips, Haplothrips aculeatus"

_insects, 2024, doi:10.3390/insects15050303_

Round 1

Reviewer 1 Report

Comments and Suggestions for Authors

Please see attached detailed report

Comments on the Quality of English Language

Author Response

  1. Reviewer #1

The manuscript reports on a novel insect-specific virus identified in H. aculeatus, an economically important pest of rice. A complete viral genome was obtained and verified using a combination of RNA sequencing, RACE technology and RT-PCR. Phylogenetic trees constructed using the viral Gp, N and RdRp proteins revealed that the virus is a member of the Aliusviridae (order Jingchuvirales) closely related to ollusvirus 1 and was provisionally named RTOV1. Finally, sRNA sequencing and analysis of vsiRNA indicated that infection of H. aculeatus is accompanied by activation of the host antiviral RNAi pathway.

The study is highly significant in that, although an olluvirus has been identified in wild bee species and grasshopper, it is the first to report on such a virus infecting Phlaeothripidae. Its characterisation therefore contributes to a further understanding of the diversity and evolution of ISVs insects which could have implications for pest management and knowledge about causes of rice damage caused by thrip species. The manuscript is recommended for publication in Insects.

Minor comments for consideration by the authors:

Given the in-depth analysis of the novel virus in an important insect pest, it is suggested that manuscript be presented as a research article rather than a brief report. Additionally, the series of experiments performed and the significant findings would be easier to follow if the format of the manuscript could be changed to include separate sections of results and discussions.

Response: We greatly appreciate your recognition of the significance of our work and your suggestion that our manuscript should be presented as a research article rather than a brief report. We have split the results and discussion sections into separate parts.

The results of the vsiRNA analysis are interesting as they indicate that the virus is probably replicating in the insect host and this has implications for pest control. They should therefore be reported briefly in the abstract. If the study (as indicated at the end of the summary) provides valuable information useful for pest management, perhaps the authors could elaborate on this in the discussion.

Response: Thanks for this reviewer’s helpful suggestions. We have already incorporated this result into the abstract (lines 31-33).

In the title, change viruses to virus.

Response: Thank you for your correction. We have rectified this mistake. (line 2)

In the Materials and Methods, there is no mention of the sample size used for RNA extraction. This is important given the small size of the insect being analysed.

Response: In the experiment, we used 50 adult thrips as a sample for RNA extraction. We have provided a comprehensive explanation for that section. (lines 77-83)

In the legend of Figure 1, ORFs were annotated in green, orange and red?

Response: In Figure 1A, predicted ORFs are labeled with light green. In Figure 1B, we distinguished proteins encoded by different ORFs using various colors: red represents RdRp, green represents N, and orange represents Gp. The two gray squares respectively indicate predicted functional domains on the RdRp protein.

Line 169: Which ORF in Figure 1 delineates the capsid region (CP) of the genome? This is not indicated in the figure or legend

Response: We apologize for the lack of clarity in our expression. In negative-sense RNA viruses, the nucleoprotein (N) is commonly referred to as the capsid protein (CP) because the N protein tightly encapsulates the viral genome within the viral particles. Therefore, we used "N" (ORF2) in the legend to delineate the capsid region.

Line 236: correct viral to virus

Response: Thank you for your correction. We have corrected this. (line 305)

Conclusion: presumably further studies will be conducted on this novel virus. Perhaps there could be some mention of where the study will go from here. For example, is any work planned to isolate the virus, determine its virulence in the host, its host range and possible transmission to rice? If, as the authors suggest, the study provides information that may be useful

Response: Thank you for your support of our work. Moving forward, we may continue to explore whether this virus causes diseases in rice and its host range.

Reviewer 2 Report

Comments and Suggestions for Authors

H. Hong et al. studied RNA from the insect Haplothrips aculeatus , a rice parasite. They identified a viral sequence of a potential negative RNA virus close to the ollusvirus group and provisionally named “ Rice thrips ollusvirus 1”. The paper is written clearly and the sequence of the new virus is far enough from the known references to be a significant contribution to the knowledge of this group of viruses. That said, the manuscript is particularly light on the methods section as well as on the presentation of the results of the small RNA library .

Lines 67-72: Too much information is missing regarding RNA sample preparation. How many insects?, what quantity of RNA?, what extraction method? With rRNA depletion or capture with oligodT and/or other…?

The abundant presence of rRNA still being a major problem in RNA extraction.

Lines 67-72: I did not find any instructions in English for the “AG RNAex pro reagent ” kit but only in Chinese. Please provide a link in English as reference to the phrase “ according to the manufacturer's instructions”.

Line 84: To my knowledge, Trinity ( https://github.com/trinityrnaseq/trinityrnaseq/releases ) has never been a trimmimg software but a transcriptome assembly software . Please check which trimming software was used on these reads .

Line 91: Since no assembly software is mentioned in this chapter, I assume that Trinity was used to produce the contigs. Thank you for clarifying this point.

Line 91-92: the phrase “contigs were used to search a local virus database …” doesn’t mean anything. You have probably used a tool to search for similarity between your contigs and the sequences in this database , but which tool? what parameters? What similarity exists between the contig(s) of interest and the sequences in the database ?

Line 127: This sequencing should be deposited in the SRA bank ( https://www.ncbi.nlm.nih.gov/sra ).

Line 160: curiously you refer to Figure 1B before referring to Figure 1A.

Figure 1B: on my PDF document it is almost impossible to tell the difference between the patterns shown in light gray and the patterns shown in dark gray.

Line 171: With which program were the reads realigned to the genome?, what parameters?

Line 175-176: Please expand your point to explain the relationship between 3'UTR coverage and efficient virus replication.

Figure 2BC: On my PDF document the bootstraps are completely unreadable. You must either increase the font size or choose a threshold and put a star each time you are above this threshold.

Figure 3A: I don't understand this figure. Most of the time the reads negative reads, representing the reverse sequence of the genome, are counted between 0 and -∞, while positive reads are counted between 0 and +∞. In your figure you have negative reads (pink) counted as positive and positive reads (cyan) counted as positive and also negative??

Line 207-230: The analysis of the small RNA library is far too simplistic. You define vsiRNAs as all the segments of your library, forgetting that this library also contains all the degradation products of viral RNA. For me, there is nothing to prove that you have siRNAs in this library, no ultra-majority peak of a certain size. Most vsiRNAs in Drosophila and mosquito ( Aedes ) have a fixed size of 21 bases. Do we have any prior information that could show that the size of haplothrips vsiRNAs would be 22 bases?

For sizes above 24 bases it is very likely that these could be vpiRNAs . For Figure 3C I advise studying the biases for 2 or 3 populations (21b, 22b and 24-28b) and also looking for a ping-pong effect proving the production of vpiRNAs ( https://www.ncbi.nlm. nih.gov/pmc/articles/PMC2995411/ Figure 3B).

Author Response

Reviewer #2

Hong et al. studied RNA from the insect Haplothrips aculeatu, a rice parasite. They identified a viral sequence of a potential negative RNA virus close to the ollusvirus group and provisionally named “Rice thrips ollusvirus 1”. The paper is written clearly and the sequence of the new virus is far enough from the known references to be a significant contribution to the knowledge of this group of viruses. That said, the manuscript is particularly light on the methods section as well as on the presentation of the results of the small RNA library.

Lines 67-72: Too much information is missing regarding RNA sample preparation. How many insects? what quantity of RNA? what extraction method? With rRNA depletion or capture with oligodT and/or other…?

The abundant presence of rRNA still being a major problem in RNA extraction.

Response: We have added relevant content in method of “Sample Preparation and RNA Extraction”. Specifically, 50 adult thrips were used as one RNA extraction sample. The RNA extraction solution used was AG RNAex Pro Reagent from Accurate Biology company, and the extraction method referenced the TRIzol Reagent (Invitrogen, Carlsbad, USA) method. 2 ug of RNA was used for reverse transcription cDNA synthesis, transcriptome sequencing, and small RNA sequencing. 500 ng of RNA was used for cDNA synthesis in RACE experiments. (lines 76-83)

During transcriptome sequencing, we opted to sequence after rRNA depletion. The data analysis was conducted based on sequencing reads devoid of rRNA.

Lines 67-72: I did not find any instructions in English for the “AG RNAex pro reagent” kit but only in Chinese. Please provide a link in English as reference to the phrase “according to the manufacturer's instructions”.

Response: The AG RNAex Pro reagent was purchased from a Chinese reagent company. Its usage method is identical to that of the TRIzol reagent from Invitrogen company. I couldn't find any relevant English version of the operating instructions online. Many articles mention the use of this reagent for RNA extraction, but none provide a description of the usage method [1,2].

Line 84: To my knowledge, Trinity ( https://github.com/trinityrnaseq/trinityrnaseq/releases ) has never been a trimmimg software but a transcriptome assembly software . Please check which trimming software was used on these reads.

Response: We employed the Trimmomatic software to remove low-quality reads and adapter sequences from sequencing data. We have made corrections. (lines 95-96)

Line 91: Since no assembly software is mentioned in this chapter, I assume that Trinity was used to produce the contigs. Thank you for clarifying this point.

Response: We apologize for any confusion caused and we will strive to improve the clarity of our descriptions.

Line 91-92: the phrase “contigs were used to search a local virus database …” doesn’t mean anything. You have probably used a tool to search for similarity between your contigs and the sequences in this database, but which tool? what parameters? What similarity exists between the contig(s) of interest and the sequences in the database?

Response: The tool used to search for similarity between contigs and sequences in the database is Diamond (with specific parameters: -e 2e-5 -f 6). After alignment, we are interested in contigs with amino acid homology lower than 90%.

Line 127: This sequencing should be deposited in the SRA bank ( https://www.ncbi.nlm.nih.gov/sra ).

Response: Thank you for this suggestion. We are currently uploading these sequences to the SRA bank. We believe that we will soon be able to showcase these data to all researchers.

Line 160: curiously you refer to Figure 1B before referring to Figure 1A.

Response: I'm not entirely clear on your specific meaning. I guess you might be thinking of one of the following two possibilities: 1. I presented the results of Figure 1B first, rather than describing them in order; 2. The references I used at Figure 1B.

If it's the first scenario, it might be because my labeling of Figure 1A wasn't clear enough, leading you to overlook that the results of Figure 1A were described before those of Figure 1B. I specifically described the results of Figure 1A from lines 151 to 162. If it's the second scenario, it's because I referenced descriptions of ollusvirus 1 (ASOV1) from other articles in Figure 1B.

Figure 1B: on my PDF document it is almost impossible to tell the difference between the patterns shown in light gray and the patterns shown in dark gray.

Response: Thank you for bringing this up. We have increased the color contrast between these two domains.

Line 171: With which program were the reads realigned to the genome? what parameters?

Response: We used the Bowtie2 software (Version 2.3.5.1) with default parameters to realign the reads to the genome.

Line 175-176: Please expand your point to explain the relationship between 3'UTR coverage and efficient virus replication.

Response: As far as I know, the 3' UTR of negative-sense RNA virus genomes is a crucial area for regulating virus replication and transcription. Transcriptome data indicates that the coverage of 3' UTR small RNAs reflects the expression level of the viral genome in host cells and the impact of RNA interference (RNAi) on virus replication. A higher coverage of small RNAs suggests a more active RNAi pathway in host cells, which can more effectively inhibit virus replication, thereby reducing virus replication efficiency. Conversely, a lower coverage of small RNAs may indicate lower activity of the RNAi pathway, making it easier for the virus to replicate and spread. Therefore, there is a close relationship between the coverage of 3' UTR small RNAs and virus replication efficiency, and transcriptome data can help reveal the interaction between them and their impact on virus biology.

Figure 2BC: On my PDF document the bootstraps are completely unreadable. You must either increase the font size or choose a threshold and put a star each time you are above this threshold.

Response: Thanks for this reviewer’s suggestions. Due to the time-consuming nature of constructing the phylogenetic tree, and the short time frame for revisions, we have chosen to enlarge Figure 2A onto one page, while enlarging Figures 2B and 2C onto another page.

Figure 3A: I don't understand this figure. Most of the time the reads negative reads, representing the reverse sequence of the genome, are counted between 0 and -∞, while positive reads are counted between 0 and +∞. In your figure you have negative reads (pink) counted as positive and positive reads (cyan) counted as positive and also negative??

Response: Sorry for causing you confusion. Upon inspection, we found an error in the output of the images. We have replaced them with the correct format.

Line 207-230: The analysis of the small RNA library is far too simplistic. You define vsiRNAs as all the segments of your library, forgetting that this library also contains all the degradation products of viral RNA. For me, there is nothing to prove that you have siRNAs in this library, no ultra-majority peak of a certain size. Most vsiRNAs in Drosophila and mosquito ( Aedes ) have a fixed size of 21 bases. Do we have any prior information that could show that the size of haplothrips vsiRNAs would be 22 bases?

Response: We also did not find any reports on the size of haplothrips vsiRNAs. However, according to reports, vsiRNAs in Frankliniella fusca (family Thripidae) are also predominantly 22 nt in size [3]. Detailed data can be found in the Supplementary Tables S1 of the reference. We have provided some descriptions regarding this in the discussion section. (lines 293-298)

For sizes above 24 bases it is very likely that these could be vpiRNAs . For Figure 3C I advise studying the biases for 2 or 3 populations (21b, 22b and 24-28b) and also looking for a ping-pong effect proving the production of vpiRNAs ( https://www.ncbi.nlm. nih.gov/pmc/articles/PMC2995411/ Figure 3B).

Response: Thank you for your suggestions. We will pay more attention to this aspect in our future research.

Reference

  1. Wang, Y., Li, J., Zheng, H. et al. Cezanne promoted autophagy through PIK3C3 stabilization and PIK3C2A transcription in lung adenocarcinoma. Cell Death Discov. 9, 302 (2023). https://doi.org/10.1038/s41420-023-01599-4
  2. Luo, C., Wang, S., Shan, W. et al. A Whole Exon Screening-Based Score Model Predicts Prognosis and Immune Checkpoint Inhibitor Therapy Effects in Low-Grade Glioma. Front Immunol. 2022 Jun 13;13:909189. doi: 10.3389/fimmu.2022.909189.
  3. Fletcher SJ, Shrestha A, Peters JR, Carroll BJ, Srinivasan R, Pappu HR, Mitter N. The Tomato Spotted Wilt Virus Genome Is Processed Differentially in its Plant Host Arachis hypogaea and its Thrips Vector Frankliniella fusca. Front Plant Sci. 2016 Sep 7;7:1349. doi: 10.3389/fpls.2016.01349.

Reviewer 3 Report

Comments and Suggestions for Authors

The authors report an interesting discovery of a new virus in a thrips.  Thrips are important pests of crops as they transmit Tospoviruses.  Biological control agents are needed for Thrips and new virus reports provide the foundational studies for the research community to continue to screen and develop use of these pathogens.  The authors report evidence that the virus is replicating in their Thrips which is an important evidential data.  While the other viruses in the genus, Ollusvirus, are a new and growing group of viruses, this provides evidence of another potential member to expand this taxa and a new report from the thips Family: Phlaeothripidae.  The grammar is good, with only a few minor corrections noted.  All scientific names should be ITALICIZED, and provided with FAMILY, ORDER, and Common name if available, the first time mentioned in the text.

Line 212 Page 7---Need Scientific names are Italicized

Line 236,  grammar,   “This virus can be….”

genus ‘o’  should be CAPITALIZED ‘Ollusvirus’

Page 159, should include the Genus Specie, then Descriptor Author (Bolivar), Family: Pyrgomorphidae, Common name: Pink-Winged Grasshopper [26].

Line 205. “The alignment analyses for RTOV1,  provided mix returns, however the strongest support was in the Family: Aliusviridae (Ollusvirus) ?? was this the case? 

I did a check and conducted blastp of the data, ORFs and got more returns (Hits) that matched more Chuviridae species, so authors should recheck and see if they should be claiming this virus to be a new member of the Taxonomic Family.  Chiviridae, that is Ollusvirus-like ?  or not.  Overall a very good report and research.

Author Response

  1. Reviewer #3

The authors report an interesting discovery of a new virus in a thrips.  Thrips are important pests of crops as they transmit Tospoviruses.  Biological control agents are needed for Thrips and new virus reports provide the foundational studies for the research community to continue to screen and develop use of these pathogens.  The authors report evidence that the virus is replicating in their Thrips which is an important evidential data.  While the other viruses in the genus, Ollusvirus, are a new and growing group of viruses, this provides evidence of another potential member to expand this taxa and a new report from the thips Family: Phlaeothripidae.  The grammar is good, with only a few minor corrections noted.  All scientific names should be ITALICIZED, and provided with FAMILY, ORDER, and Common name if available, the first time mentioned in the text.

Response: Thank you for your careful reading of the manuscript. We have corrected these errors.

Line 212 Page 7---Need Scientific names are Italicized

Response: We have corrected it. (line 228)

Line 236,  grammar,   “This virus can be….”

genus ‘o’  should be CAPITALIZED ‘Ollusvirus’

Response: Thank you for pointing that out. We need to enhance our language proficiency further. We have also made the correction. (line 304, line 306)

Page 159, should include the Genus Specie, then Descriptor Author (Bolivar), Family: Pyrgomorphidae, Common name: Pink-Winged Grasshopper [26].

Response: We have supplemented these contents. (lines 173-174)

Line 205. “The alignment analyses for RTOV1, provided mix returns, however the strongest support was in the Family: Aliusviridae (Ollusvirus) ?? was this the case? 

Response: Because the RdRp protein of the virus tends to be relatively conserved during evolution, considering this characteristic, we focus on the evolutionary relationship of RdRp as a classification criterion. In this paper, regardless of whether RdRp is examined through BLASTp or phylogenetic tree analysis, it shows close affinity to the genus Ollusvirus. Additionally, another point is that the length and structure of the virus genome also align more closely with those of the Ollusvirus genus.

I did a check and conducted blastp of the data, ORFs and got more returns (Hits) that matched more Chuviridae species, so authors should recheck and see if they should be claiming this virus to be a new member of the Taxonomic Family.  Chiviridae, that is Ollusvirus-like?  or not. Overall a very good report and research.

Response: We also conducted a BLASTp search for RdRp. The results show that, except our own submitted virus sequence, 9 out of the top 10 highly homologous virus sequences belong to the genus Ollusvirus. Although they appear to be Chuviridae based on their names, according to the report by Di Paola N et al (Fig 3), they are all ollusviruses [Di Paola N et al., 2022].

Reference:

   Di Paola N, Dheilly NM, Junglen S, Paraskevopoulou S, Postler TS, Shi M, Kuhn JH. Jingchuvirales: a New Taxonomical Framework for a Rapidly Expanding Order of Unusual Monjiviricete Viruses Broadly Distributed among Arthropod Subphyla. Appl Environ Microbiol. 2022 Mar 22;88(6):e0195421. doi: 10.1128/AEM.01954-21.

Reviewer 4 Report

Comments and Suggestions for Authors

Characterization of A Novel Insect-Specific Viruses Discovered  in the Rice Thrips, Haplothrips aculeatus.

Hao Hong et al

Insects

The authors present a manuscript that shows the identification of a novel negative stranded viruses infecting Rice Thrips. The authors present the ork in a concise and logical manner using appropriate technologies.

There are several English syntax and editing errors that need to be addressed through out the manuscript. An example are outline below. Several instance where mor information would make the manuscript more informative are also outlined.

Recommendation - minor revision.

Title suggests plural ‘viruses…’ should be singular ‘virus’

Line 13 suggest plural ‘…full genomes…’ and viruses discovered singular

Line 16 syntax  ‘..this viral was..’ should be ‘…this virus was…

Line 23 syntax ‘..are one of the

Line 38 – the introduction is a little light on back ground information. A concluding paragraph of the findings can be added.

Line 49 – rewrite - ‘Plant- and insect-infecting NSVs have increased in number’  In what context have they increased in number? I suspect the detection of NSVs is increasing due to the increasing transcriptome and genetic information available. Not that the NSVs have increased in number per see.

Line 52: Syntax – More information decription needed. It exhibits a wide variety of genome organisations ‘.   What is it? Eg Members of the family exhibit a wide….’ The viral genomes ‘generally’ encode a glycoprotein (G), a nucleoprotein (N), and a polymerase (L), while some viral genomes ‘within the family’ lack the glycoprotein.

Line 67 : Remove ‘The’

Line 263 : syntax – change ‘viral’  to ‘virus’

Comments on the Quality of English Language

As above

Author Response

  1. Reviewer #4

The authors present a manuscript that shows the identification of a novel negative stranded viruses infecting Rice Thrips. The authors present the ork in a concise and logical manner using appropriate technologies.

There are several English syntax and editing errors that need to be addressed through out the manuscript. An example are outline below. Several instance where mor information would make the manuscript more informative are also outlined.

Recommendation - minor revision.

Title suggests plural ‘viruses…’ should be singular ‘virus’

Response: It's our fault. We have corrected it. (line 2)

Line 13 suggest plural ‘…full genomes…’ and viruses discovered singular

Response: Sorry for this mistake. We need to enhance our proficiency in English writing. We have made the necessary corrections. (lines 13-14)

Line 16 syntax  ‘..this viral was..’ should be ‘…this virus was…

Response: We have corrected it and rephrased it. (line 16)

Line 23 syntax ‘..are one of the

Response: Thank you for pointing that out. We have made the necessary changes and will be more mindful of singular and plural usage in the future. (line 23)

Line 38 – the introduction is a little light on back ground information. A concluding paragraph of the findings can be added.

Response: Thank you for your helpful suggestions. We have added some summary descriptions of the results in introduction. (lines 68-72)

Line 49 – rewrite - ‘Plant- and insect-infecting NSVs have increased in number’ In what context have they increased in number? I suspect the detection of NSVs is increasing due to the increasing transcriptome and genetic information available. Not that the NSVs have increased in number per see.

Response: We apologize for the inaccuracy in that description. We also believe that with the advancement of detection technology, the number of known plant and insect-infecting NSVs is increasing. We have rephrased accordingly. (lines 50-52)

Line 52: Syntax – More information decription needed. It exhibits a wide variety of genome organisations ‘.   What is it? Eg Members of the family exhibit a wide….’ The viral genomes ‘generally’ encode a glycoprotein (G), a nucleoprotein (N), and a polymerase (L), while some viral genomes ‘within the family’ lack the glycoprotein.

Response: Here, 'it' refers to the order Jingchuvirale. We have made the change. (line 54)

Line 67 : Remove ‘The’

Response: We have removed it. (line 75)

Line 263 : syntax – change ‘viral’  to ‘virus’

Response: Thank you for pointing that out. We need to enhance our language proficiency further. We have also made the correction. (line 304)